# Restructuring personal networks with a Motivational Interviewing social network intervention to assist the transition out of homelessness: A randomized control pilot study

**David P. Kennedy** * , **Karen Chan Osilla, Sarah B. Hunter, Daniela Golinelli, Ervant Maksabedian Hernandez, Joan S. Tucker**

RAND Corporation, Santa Monica, California, United States of America

* davidk@rand.org

## Abstract

**Data Availability Statement:** All data file are available in Github repository: https://github.com/

### Background

Social relationships play a key role in both substance use and homelessness. Transitioning out of homelessness often requires reduction in substance use as well as changes in social networks. A social network-based behavior change intervention that targets changes personal social networks may assist the transition out of homelessness. Most behavior change interventions that incorporate social networks assume a static network. However, people experiencing homelessness who transition into housing programs that use a harm reduction approach experience many changes in their social networks during this transition. Changes may include disconnecting from street-based network contacts, re-connecting with former network contacts, and exposure to new network members who actively engage in substance use. An intervention that helps people transitioning out of homelessness make positive alterations to their social networks may compliment traditional harm reduction housing program services.

### Methods

We conducted a pilot randomized controlled trial (RCT) of an innovative Social Network Intervention (MI-SNI), which combines network visualization and Motivational Interviewing to assist adults transitioning out of homelessness. The MI-SNI provides feedback to new residents about their social environments and is designed to motivate residents to make positive changes in both their individual behavior and their personal network. In a sample of 41 adult housing program residents with past year risky substance use, we examined whether participants randomized to receive a MI-SNI showed greater changes in their personal networks over 3 months compared to those receiving usual care.

qualintitative/EgoWeb-Project-Data/tree/main/
PONE-D-19-36073R1.

**Funding:** This work is supported by grant R34
DA034855 from the National Institute on Drug
Abuse (NIDA). The funder provided support in the
form of salaries for the authors, but did not have
any additional role in the study design, data
collection and analysis, decision to publish, or
preparation of the manuscript. The specific roles of
these authors are articulated in the 'author
contributions' section.

**Competing interests:** The authors have declared
that no competing interests exist.

## Results

There were significant differences in the networks of the MI-SNI group compared to the group receiving usual care at follow-up, controlling for baseline network characteristics. The MI-SNI group had greater reductions in the proportion of their network members who influenced alcohol or other drug use (AOD) use, such as drinking partners, and more frequently changed their relationships in the direction of lower AOD risk with network members who were retained in their networks across waves.

## Conclusions

This study is the first pilot test of a MI-SNI customized for assisting the transition out of homelessness to test for personal network changes. Results indicate that MI-SNIs can have a positive impact on short-term network changes and thus may serve as a useful adjunct to behavioral change interventions. These findings suggest that a MI-SNI approach may help individuals experiencing homelessness and risky AOD use positively restructure their social networks while transitioning into supportive housing. These promising results suggest the need for a larger RCT test of this innovative intervention approach.

## Trial registration

ClinicalTrials.gov Identifier: NCT02140359.

## Introduction

Social relationships play a key role in a variety of public health problems [1–3], including alcohol and other drug (AOD) use and homelessness [4–6]. AOD use spreads through networks [7] due to a variety of network mechanisms, such as social comparison, social sanctions and rewards, flows of information, support and resources, stress reduction, and socialization [8–10]. Homelessness is often precipitated by AOD use problems [11] and continued AOD use among people experiencing homelessness is influenced by continued exposure to AOD use in their social networks [12–15]. Continued AOD use impedes transitioning out of homelessness and into housing assistance, such as when AOD abstinence is a requirement for housing. Therefore, addressing the interrelated problems of AOD use and homelessness requires a focus on social networks, which play a wide range of positive and negative roles in assisting and impeding the transition out of homelessness [12–14,16–26].

Many behavior change interventions informed by social network analysis (SNA) have been developed recently [27–30], have addressed AOD use in a variety of populations [30], and can potentially address AOD use among people experiencing homelessness. Four styles of incorporating networks into interventions have emerged [27]: 1) identifying groups in a network to target based on structural position ("segmentation"), 2) identifying and intervening with key individuals based on their structural location ("opinion leaders"), 3) activation of new interactions between people without existing ties in a network ("induction"), and 4) changing the existing network ("alteration"). For the most part, network intervention approaches use methods informed by diffusion of innovation theory [31] and aim to maximize the effects of a behavior change intervention through its spread within a well-defined and clearly bounded network (such as students in the same school).

There are challenges in applying diffusion-based SNA behavior interventions to assist people to reduce AOD use while transitioning out of homelessness. The segmentation, opinion leader, and induction approaches are inappropriate because they assume a static, bounded network [27]. However, people transitioning out of homelessness and into housing programs do not belong to a clearly defined network. They can experience heightened social volatility due to loss of contact with people they interacted with on the street, coupled with sudden and ongoing contact with new neighbors. Distancing themselves from AOD using network members may help them decrease AOD use by reducing exposure to high-risk behavior. At the same time, these individuals may have developed strong and supportive ties while living on the street and may have reservations about ending these relationships, even with members of their network who they realize hamper their efforts at positive behavior change and stability. Those who transition into housing programs may experience increased opportunities to develop new pro-social connections and reconnect with positive network ties who can provide key social support necessary to reduce AOD use. On the other hand, transitioning into a housing program that uses a harm reduction model [15,26,32,33] may result in continued exposure to AOD because these programs do not require residents to abstain from AOD use. This social upheaval experienced by individuals transitioning out of homelessness suggests that an AOD reduction behavior change intervention informed by SNA that assumes a static and bounded network is inappropriate. Network "alteration" intervention approaches, on the other hand, do not make this assumption and appear to be a better fit for addressing the social volatility associated with transitioning out of homelessness.

Our team recently developed a Motivational Interviewing Social Network Intervention (MI-SNI) designed to reduce AOD use among adults with past year problematic AOD use who recently transitioned from being homelessness to residing in a housing program [34–36]. The MI-SNI targets alterations of the "personal" networks of independently sampled individuals, rather than individuals who are members of a static, bounded network [37–39]). This approach is appropriate for people transitioning out of homelessness because each person experiencing this transition is at the center of a unique and evolving group of interconnected people who play a variety of roles in assisting or hampering their transition. The MI-SNI combines visualizations of personal network data with Motivational Interviewing (MI), an evidence-based style of intervention delivery that triggers behavior change through increased self-determination and self-efficacy while reducing psychological reactance [40,41]. Results from a pilot randomized controlled trial of the MI-SNI on AOD-related outcomes found that formerly homeless adults who recently transitioned to a housing program and received the intervention experienced reductions in AOD use, and increased AOD readiness to change and abstinence self-efficacy, compared to those who were randomly assigned to the control condition [34]. Examining whether the MI-SNI is associated with actual changes to participants' social networks, a hypothesized mechanism through which it is expected to affect AOD-related outcomes, is an important next step in this line of research.

The present study compares personal network composition and structure data collected before and after the intervention period to explore if the MI-SNI was associated with longitudinal changes in personal networks of MI-SNI intervention participants compared to participants who received usual case management services. This study provides a preliminary test of several hypotheses. Our primary hypothesis was that the intervention would be associated with a change in network composition, primarily a decrease in number network members who influence the participants' AOD use, such as those who are drinking or drug use partners. We also hypothesized that receiving visualization feedback that highlighted supportive network members would prompt participants to take steps to retain supportive network members, drop unsupportive members of their networks, and add new network members who provide

support, leading to an overall increase in supportive ties. For alters who remained in the network after the intervention period, we hypothesized that MI-SNI recipients would be more likely to change their relationships with these network members, resulting in fewer AOD risk behaviors with them. Finally, we tested an exploratory hypothesis that MI-SNI recipients would make changes to their networks that would result in them having significantly different overall network structures (size and connectivity among network members) and more network member turn-over between waves. Finding such intervention effects on network structure and turn-over would suggest that the MI-SNI influenced how participants interacted with their social environments during the intervention period.

## Material and methods

### Intervention design, setting, and participants

The complete and detailed plan for the conduct and analysis of this Stage 1a-1b randomized controlled trial (RCT) is available elsewhere [36] and the clinical trial has been registered (ClinicalTrials.gov Identifier: NCT02140359). Detailed descriptions of the development and beta testing of the Stage 1a computer interface, feasibility tests of the intervention procedures, pilot test participant characteristics, and initial pilot test results are also available elsewhere [35,36]. Participants were new residents of a housing program for adults transitioning out of homelessness in Los Angeles County recruited between May 2015 and August 2016. The primary analytic sample comes from the initial pilot test site, Skid Row Housing Trust (SRHT), which provides Permanent Supportive Housing (PSH) [42–47] services in Skid Row, Los Angeles. PSH programs do not require AOD abstinence or treatment, but do provide case management and other supportive services such as mental health and substance abuse treatment. SRHT residents and staff participated in project planning and beta testing prior to recruitment [35,36]. The intervention procedures were designed to be delivered during typical case manager sessions with new residents to supplement and improve the support they provide residents by raising both the case manager's and resident's awareness of the role that the resident's social environment plays in the transition out of homelessness. Beginning in February 2016, an additional supplemental sample was recruited from SRO Housing Corporation (SRO Housing), which is a similar housing program also located in Skid Row. This additional recruitment was in response to slower than expected monthly recruitment rates from SRHT and a projected shortfall in our targeted recruitment sample size of 15–30 subjects per intervention arm, which is a rule-of-thumb recommendation of the National Institutes of Drug Abuse for funding Stage 1b Pilot Trials [48].

Residents were recruited through SRHT and SRO Housing leasing offices prior to receiving the assignment of a housing unit. Eligible participants were > = 18 years old English speakers who had been housed within the past month and were screened for past-year harmful alcohol use (Alcohol Use Disorders Identification Test (AUDIT-C) score > = 4 for men and > = 3 for women) [49] or drug use (Drug Abuse Screen Test (DAST) score greater than 2) [50–52]. The 149 residents who were contacted by the research team resulted in a 49 eligible residents who were randomized into the intervention arm (N = 25) or the control arm (N = 24) using a permuted block randomization strategy stratified by gender. Full recruitment details and results are provided in the Fig 1 CONSORT diagram along with a CONSORT checklist in S1 Appendix. Eligible residents were informed of their rights as research participants and provided written consent. Retention in the study was excellent, with 84% of participants (n = 21 intervention, n = 20 control) completing the follow-up assessment three months later. Participants averaged 48 years of age, were primarily male (80%), African American (56%), had a high school education or less (68%), were never married (66%), had children (59%), and

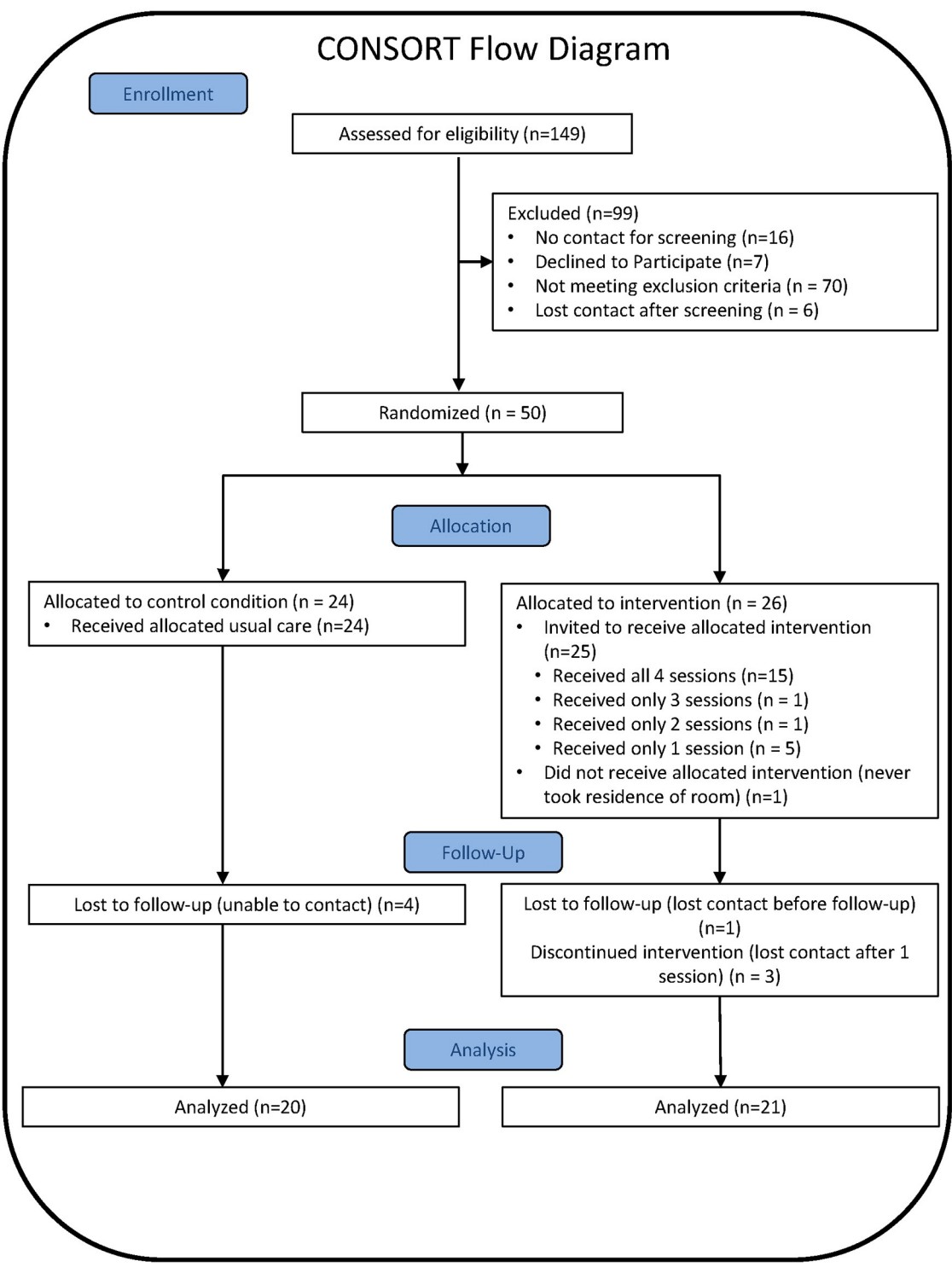

**Fig 1. CONSORT diagram of pilot study recruitment.**

received an average of $471 in monthly income. Full details about participant demographics and AOD use is available elsewhere [34]. All procedures were approved by the authors' Institutional Review Board (IRB) (Study ID: 2013-0373-CR02) and the complete and detailed plan for the conduct and analysis of the trial that was approved by the IRB before the trial began is available in S2 Appendix. A Federal Certificate of Confidentiality was obtained for this study, which provided additional privacy protection from legal requests.

## Baseline and follow-up data collection procedures

The purpose of the baseline and follow-up network data collection assessments was to measure participants' personal network characteristics when they first moved into their supportive housing unit and 3 months later to provide measures of network change and test if those who were offered the intervention experienced significantly different network changes compared to those randomly assigned to the control condition. Personal network assessment interviews were conducted through one-on-one, in-person interviews (~45–60 minutes) by independent data collectors who did not have access to the assignment of IDs to study arm and were therefore blind to study condition. Interviews were conducted using the social network data collection software EgoWeb 2.0 (egoweb.info) installed on a laptop computer. Participants were paid $30 to complete the baseline interview and $40 for the follow-up.

We followed common procedures for collecting personal network data [37] used in previous studies of AOD use and risky sex among homeless populations [12–14,53–57]. Respondents (referred to as the "egos" in personal network interviews) were first asked questions about themselves, including demographic questions (baseline only) and a series of questions about their own AOD use. After these questions, the egos were asked the following standard question prompting them to name up to 20 people in their network (referred to as a "name generator" question):

"Now I'd like to ask you some questions about the people that you know. First, I'd like for you to name 20 adults, over 18 years old, that have been involved in your life over the past year. We do not want their full names—you can use their first names, initials or descriptions. These should be people you have had contact with sometime in the past year–either face-to-face, by phone, mail, e-mail, text messaging, or online. Start by naming the people who have been the most significant to your life—either in a positive way or a negative way. You can decide for yourself who has been significant, but consider those who have had a significant emotional, social, financial, or any other influential impact on your life. We'll work outwards toward people who have less significance. You can name any adults you have interacted with no matter who they are or where they live or how much time you have spent with them."

Once each ego provided a list of names (referred to as network "alters"), they were asked a series of questions about each person (referred to as "name interpreter" questions). Respondents were also asked, for each unique alter-alter tie, if these two people knew each other. These personal network questions were asked at both baseline and follow-up and these responses provided raw data for measurements of change in personal networks.

## Intervention procedures

Residents who completed baseline interviews were randomly assigned to either the intervention or control arms. Those assigned to the intervention arm were offered four biweekly in-person sessions with a MI-trained facilitator. Full details about the intervention delivery, including examples of the visualizations presented to participants during the session, are available elsewhere [34]. Briefly, facilitators conducted a brief personal network interview (~15 minutes) focusing on recent network interactions (past 2 weeks). Name generator and name

interpreter question wording were selected to generate a series of visualizations of the resident's recent interactions with their immediate social network. These visualizations highlighted different aspects of the network (network centrality, AOD use, social support) and were used to guide a conversation about the participant's social network in a MI session that immediately followed the personal network interview.

## Measures

**Network outcomes: AOD use/influence.** We constructed four types of network AOD use/influence measures from three name interpreter questions. Participants identified which alters they drank alcohol with and whether they engaged in this behavior over the past 4 weeks. Based on this question, alters were categorized as a **drinking partner** and a **recent drinking partner**. A similar question was asked about other drug use with each alter, which was used to classify alters as a **drug use partner** and **recent drug use partner**. Participants were also asked if they drank more alcohol or used more drugs than usual when they were with the alter and if this happened recently. This question was used to classify each alter as an **AOD use influence** alter and a **recent AOD use influence** alter. These variables were combined to produce overall **any risk** or **any recent risk** dichotomous variables if any of the above variables was true. For each of these dichotomous variables, an overall network proportion variable was constructed by summing the number of alters with the characteristic and dividing by the total number of alters in the ego's personal network.

**Network outcomes: Social support.** We constructed four types of network social support variables from 3 name interpreter questions. Respondents were asked if they received three different types of support from each alter: **emotional support** (e.g. encouragement), **information support** (e.g. advice), and **tangible support** (e.g. money, transportation, food) and if this support happened in the past 4 weeks. Alters were classified as having given each of these types of support both ever and recently. Also, alters who provided at least one of these types of support were classified as **any support** and **any recent support**. For each alter social support variable, an overall network support proportion variable was constructed by summing the number of alters with the support characteristic and dividing by the total number of alters in the ego's personal network.

**AOD risk relationship change outcomes.** We constructed four types of AOD risk relationship measures to test if the intervention was associated with egos changing their AOD related behavior with alters who remained in their networks (in contrast to alters who were removed or added to their networks across assessments). To construct these measures, we first identified which network members were named at both assessments by matching the names listed at the baseline and follow-up interview for each respondent to identify unique alters. Next, we compared the responses about retained alters' AOD risk at the baseline and follow-up assessments to identify those who changed status as drinking partners, drug use partners, AOD use influence partners, or any AOD risk partners. For each of these four types of status changes, we constructed: (a) **stopping** measures to indicate alters who had the characteristic at baseline, but not have it at follow-up; and (b) **starting** measures to indicate alters who did not have the characteristic at baseline, but had it at follow-up. We constructed overall measures of each of these variables for each ego by counting the number of alters who had the relationship change characteristic.

**Network structure and network member turnover outcomes.** We constructed measures of overall cross-wave network structure to explore associations with overall network size and interconnectivity intervention status. Matching alter names across waves enabled construction of a cross-wave network that included all alters named at either wave. Next, based on these

cross-wave networks, we constructed common measures of personal network structure [38] including a measure of network size (i.e., **total unique alters named**), and two measures of network connectivity: **cross-wave density** (the ratio of existing ties between network members to the total possible number of ties) and **cross-wave components** (number of groups of network members with no connections to other members of the network). To measure network turnover, alters who were named in only one wave were classified as either **dropped alters** (baseline only), **added alters** (follow-up only), or **retained alters** (named in both waves). For each respondent, we constructed counts of dropped, added, or retained alters in the cross-wave network.

**Background variables.** Demographic and AOD use variables were used to inform the construction of model weights to adjust for participants who did not complete both assessments. Demographic variables captured in the baseline assessment included age, gender, race/ethnicity, education, number of children, marital status, and income. AOD use variables included the quantity and frequency of alcohol use and days using marijuana in the past 4 weeks and an assessment of readiness to change AOD use [58]. Also included in the construction of weights were variables assessing housing program (SRHT vs. SRO) and intervention arm.

## Analyses

The primary goal of the current study is to provide preliminary empirical evidence of the intervention's effect on changes to network composition and structure for intervention recipients compared to participants assigned to the control condition. The results presented here were generated with the same analytic approach as a previous study that found an association between receipt of the intervention and changes in participants' AOD behavior and attitudes [34]. We used an intent-to-treat [59] approach by offering follow-up to all participants and analyzing their data to reduce type I errors [60]. We constructed and tested a series of regression models with each AOD use/influence proportion and each social support proportion from the follow-up assessment as the dependent variable with the intervention group indicator as the predictor variable, while controlling for the baseline measure of the dependent variable. We also constructed regression models with each of the AOD risk relationship count variables and each network structure and turnover variable with intervention group as predictor variable while controlling for network size at baseline. We used linear regression for continuous outcomes and Poisson regression for count outcomes.

The models were fitted using the "survey" package in R version 3.3.1 to include non-response weights. These weights enabled computation of accurate standard errors and accounting for the potential bias caused by unit non-response missing data [61] due to participants skipping the follow-up assessment or dropping out of the study. Of the 49 eligible study participants who completed a baseline assessment, 41 also completed the 3-month follow-up assessment and responders differed from non-responders on a few characteristics, such as income and whether they were housed in SRHT or SRO Housing. The nonresponse weights were estimated using a non-parametric regression technique, called boosting [62,63], instead of logistic regression, as implemented in the TWANG R package [64] and including baseline outcome and demographic variables in the model. We calculated effect sizes based on parameter estimates from the regressions and pooled standard deviation at baseline to calculate Cohen's *d* [65,66].

For each model, we conducted two stages of analysis, similar to our previous approach [34]. First, we analyzed data from our primary sample, the 28 participants from SRHT only, because the MI-SNI was developed for SRHT residents with input from SRHT staff and SRHT case

management process. Second, we conducted a secondary analysis on the full sample of 41 that included the small number of residents from a different housing program (SRO Housing). Additional details about the justification of this two stage approach, including details about the differences between the participant samples, is available elsewhere [34].

## Results

### Network composition

Table 1a presents descriptive statistics (i.e., mean and standard deviations) for the baseline and follow-up network proportion measures for the SRHT intervention and control group participants. In addition, the table presents results from the regression models for the SRHT sample analysis predicting the intervention effect on proportions of types of network members at follow-up controlling for the same baseline network composition measures. Each row presents the results of one model. Table 1b presents these same findings for the full sample. The intervention effect was significant at the 95% confidence level with a large effect size for the proportion of drinking partners in the network at follow-up for the SRHT residents only. On average, intervention recipients had 13% fewer recent drinking partners in their networks at follow-up compared to participants in the control arm, controlling for baseline personal network composition (p = .042, $d$ = .81). The average change in proportion of recent drinking partners in the overall sample was not significantly different between intervention and control recipients (p = .145). There was also a significant decrease in proportion of alters with "any risk" for the SRHT residents only. The average 13% decrease in the SRHT-only sample for intervention participants compared to control participants was marginally significant with a medium to large effect size (p = .063, $d$ = .74). The model for the full sample did not reach significance (p = .21).

Table 2a (SRHT only) and 2b (full sample) provide descriptive statistics for counts of alters who changed their AOD use and risk influence relationship status with egos between baseline and follow-up assessments and results of models testing if intervention status was significantly associated with these counts. Each model with count outcomes controlled for size of the network at baseline (number of alters named). The tables also present results of exploratory tests of the intervention effects on network structure and turnover. Each model estimate and 95% CIs were converted to incident rate ratios (IRR) (excluding network density) because model estimates of count outcomes can be easily interpreted as predicted % increase or decrease [67].

Models testing for associations between intervention arm and counts of changing relationships identified several medium-sized effects for the full sample. First, intervention participants had an average of 2.68 times more retained alters who stopped being drinking partners (i.e., the respondent reported as a drinking partner at baseline but not at follow-up) compared to control participants (p = .03, $d$ = .61). Second, when considering those who influenced AOD use with respondents at follow-up but not baseline (i.e., classified as starting AOD use influence), intervention participants had only 13% the number of these retained alters in their networks compared to similar control participants (p = .02, $d$ = .59). Third, intervention recipients had an average of 42% fewer retained alters who changed from not being rated as having any of the three risk characteristics at baseline to having at least one at follow-up compared to control participants (p = .05, $d$ = .52). These associations were not significant within the SRHT-only sample (see Table 2a) except for a marginally significant decrease in alters who started influencing AOD use between waves: SRHT intervention participants averaged only 10% the number of these retained alters in their networks compared to similar control participants (p = .07, $d$ = .49).

**Table 1.** a. Regression model results for intervention effects on personal network composition, SRHT only (N = 28). b. Regression model results for intervention effects on personal network composition for full sample (N = 41).

| Proportion of alter types | Control (N = 13) | | Intervention (N = 15) | | Est.[2] | p-val | 95% CI[2] | d[3] | Control (N = 20) | | Intervention (N = 21) | | Est.[2] | p-val | 95% CI[2] | d[3] |
|---|---|---|---|---|---|---|---|---|---|---|---|---|---|---|---|---|
| | Baseline | Follow-up | Baseline | Follow-up | | | | | Baseline | Follow-up | Baseline | Follow-up | | | | |
| | mean (SD)[1] | mean (SD) | mean (SD) | mean (SD) | | | | | mean (SD)[1] | mean (SD) | mean (SD) | mean (SD) | | | | |
| Drinking Partners | .24 (.17) | .30 (.27) | .28 (.33) | .22 (.18) | -.09 | .320 | (-.25, .08) | .37 | .21 (.20) | .27 (.25) | .32 (.34) | .21 (.20) | -.06 | .342 | (-.19, .06) | .46 |
| Recent Drinking Partners | .09 (.13) | .20 (.26) | .13 (.16) | .07 (.14) | -.13 | **.042** | (-.26, -.01) | .81 | .08 (.12) | .14 (.22) | .16 (.23) | .06 (.12) | -.08 | .145 | (-.19, .03) | .63 |
| Drug Use Partners | .13 (.13) | .17 (.19) | .22 (.25) | .12 (.14) | -.06 | .379 | (-.18, .07) | .56 | .14 (.18) | .15 (.17) | .28 (.28) | .12 (.15) | -.02 | .598 | (-.11, .06) | .54 |
| Recent Drug Use Partners | .02 (.04) | .07 (.11) | .06 (.09) | .06 (.12) | -.01 | .744 | (-.10, .07) | .38 | .03 (.06) | .05 (.09) | .11 (.19) | .04 (.10) | -.01 | .740 | (-.08, .05) | .54 |
| AOD Use Influence | .10 (.14) | .16 (.19) | .11 (.15) | .11 (.18) | -.05 | .456 | (-.18, .08) | .31 | .12 (.19) | .14 (.17) | .17 (.24) | .09 (.15) | -.05 | .270 | (-.15, .04) | .40 |
| Recent AOD Use Influence | .03 (.09) | .04 (.07) | .04 (.10) | .03 (.09) | -.01 | .613 | (-.07, .04) | .20 | .03 (.08) | .03 (.06) | .10 (.21) | .02 (.07) | -.01 | .695 | (-.05, .03) | .44 |
| Any Risk | .28 (.21) | .33 (.27) | .35 (.31) | .24 (.21) | -.09 | .287 | (-.26, .07) | .48 | .29 (.25) | .31 (.25) | .40 (.32) | .25 (.22) | -.06 | .364 | (-.18, .07) | .47 |
| Recent Any Risk | .10 (.12) | .22 (.25) | .15 (.16) | .09 (.17) | -.13 | **.063** | (-.26, .00) | .74 | .10 (.12) | .15 (.22) | .21 (.25) | .08 (.15) | -.07 | .206 | (-.19, .04) | .64 |
| Emotional Support | .77 (.22) | .88 (.24) | .80 (.22) | .82 (.25) | -.06 | .490 | (-.22, .11) | .27 | .76 (.22) | .77 (.30) | .79 (.20) | .81 (.22) | .04 | .596 | (-.11, .19) | .07 |
| Recent Emotional Support | .60 (.25) | .69 (.33) | .66 (.29) | .51 (.30) | -.18 | .127 | (-.40, .04) | .58 | .56 (.28) | .60 (.35) | .64 (.25) | .47 (.28) | -.13 | .131 | (-.30, .04) | .52 |
| Informational Support | .78 (.30) | .76 (.35) | .77 (.27) | .86 (.17) | .10 | .333 | (-.10, .30) | .33 | .76 (.27) | .70 (.34) | .74 (.28) | .81 (.17) | .11 | .170 | (-.04, .26) | .39 |
| Recent Informational Support | .61 (.26) | .61 (.34) | .57 (.35) | .59 (.33) | -.02 | .852 | (-.25, .21) | .00 | .57 (.28) | .53 (.34) | .54 (.31) | .53 (.30) | .00 | .979 | (-.17, .18) | .07 |
| Tangible Support | .55 (.33) | .56 (.44) | .59 (.35) | .57 (.37) | .01 | .945 | (-.25, .27) | .02 | .55 (.33) | .48 (.42) | .61 (.31) | .49 (.35) | .02 | .877 | (-.20, .23) | .04 |
| Recent Tangible Support | .35 (.22) | .18 (.17) | .31 (.26) | .16 (.17) | -.03 | .642 | (-.14, .09) | .02 | .32 (.27) | .19 (.24) | .30 (.26) | .14 (.15) | -.04 | .409 | (-.15, .06) | .14 |
| Any Support | .85 (.24) | .92 (.24) | .89 (.16) | .92 (.10) | .00 | .971 | (-.10, .10) | .11 | .82 (.22) | .84 (.28) | .89 (.15) | .88 (.11) | .05 | .382 | (-.06, .15) | .06 |
| Recent Any Support | .66 (.27) | .74 (.29) | .70 (.30) | .63 (.30) | -.11 | .296 | (-.32, .09) | .43 | .63 (.27) | .65 (.32) | .69 (.27) | .57 (.29) | -.08 | .330 | (-.24, .08) | .37 |

[1]Baseline and Follow-up means and SDs weighted from full intent-to-treat sample (N = 49) to account for non-response at follow-up.

[2]Weighted intervention effect estimates and 95% Confidence Intervals from linear regression models predicting follow-up measure controlling for baseline.

[3]Cohen's d effect sizes interpreted as small (.20), medium (.50), and large (.80).

## Network structure and turn-over

For overall network structure, several significant effects of medium-to-large magnitude were found. Average cross-wave network density was 0.18 higher for intervention participants compared to control participants in the SRHT-only sample (p = .02, d = .82), although this association was not significant in the full sample (p = .14). For cross-wave network number of components, intervention networks had on average 55% the number of components as the control arm for the full sample (p = .02, d = .74) and 42% for the SRHT-only sample (p < .01, d = 1.01). Overall network size did not significantly differ between treatment conditions.

**Table 2.** a. Regression model results for intervention effects on relationship change count outcomes, network structure, and network turnover, SRHT only (N = 28). b. Regression model results for intervention effects on relationship change count outcomes, network structure, and network turnover, full sample (N = 41).

| | Control (N = 13) | Intervention (N = 15) | Est.[2] | p-val | 95% CI[2] | d[3] | Control (N = 20) | Intervention (N = 21) | Est.[2] | p-val | 95% CI[2] | d[3] |
|---|---|---|---|---|---|---|---|---|---|---|---|---|
| | mean(SD)[1] | mean(SD) | | | | | mean(SD)[1] | mean(SD) | | | | |
| *Relationship Change* | | | | | | | | | | | | |
| *Stopping behavior with alter* | | | | | | | | | | | | |
| Drinking with | .91 (1.61) | 1.97 (2.66) | 2.39 | .22 | (.62, 9.24) | .47 | 1.18 (1.86) | 3.24 (4.18) | 2.68 | **.03** | (1.12, 6.42) | .61 |
| Using Drugs With | .68 (1.10) | 1.58 (2.34) | 2.61 | .16 | (.71, 9.58) | .47 | 1.32 (3.32) | 2.89 (3.83) | 2.03 | .28 | (.57, 7.21) | .43 |
| Influenced to use AOD | .45 (.87) | .90 (1.27) | 2.09 | .30 | (.53, 8.20) | .41 | 1.17 (3.12) | 2.08 (3.55) | 1.58 | .52 | (.40, 6.24) | .27 |
| Any AOD risk | 1.14 (1.87) | 2.30 (2.55) | 2.11 | .22 | (.66, 6.79) | .50 | 2.11 (3.93) | 3.82 (4.45) | 1.69 | .30 | (.64, 4.47) | .40 |
| *Starting behavior with alter* | | | | | | | | | | | | |
| Drinking with | .85 (.98) | .80 (1.54) | .84 | .74 | (.29, 2.38) | .04 | .90 (.91) | .84 (1.43) | .80 | .59 | (.37, 1.76) | .06 |
| Using Drugs With | .56 (.88) | .25 (.58) | .44 | .26 | (.11, 1.80) | .41 | .41 (.76) | .27 (.55) | .60 | .40 | (.19, 1.92) | .22 |
| Influence to use AOD | .64 (1.68) | .07 (.26) | .10 | **.07** | (.01, 1.13) | .49 | .71 (1.43) | .09 (.30) | .13 | **.02** | (.02, .67) | .59 |
| Any AOD risk | .93 (1.11) | .39 (.81) | .38 | .12 | (.12, 1.23) | .55 | .91 (1.07) | .43 (.74) | .42 | **.05** | (.18, .99) | .52 |
| *Cross-wave network structure* | | | | | | | | | | | | |
| Total Unique Alters Named | 24.64 (9.38) | 23.93 (8.00) | .97 | .83 | (.75, 1.26) | .08 | 23.96 (9.10) | 23.71 (7.54) | .99 | .92 | (.80, 1.22) | .03 |
| Cross-Wave Density[2] | .19 (.14) | .36 (.24) | .18 | **.02** | (.03, .32) | .82 | .23 (.18) | .32 (.22) | .09 | .14 | (-.03, .21) | .45 |
| Cross-Wave Components | 7.72 (5.36) | 3.21 (2.09) | .42 | **< .01** | (.25, .69) | 1.01 | 6.58 (5.04) | 3.60 (2.02) | .55 | **< .01** | (.36, .83) | .74 |
| *Alter turnover* | | | | | | | | | | | | |
| Dropped Alters | 9.52 (4.85) | 7.48 (5.52) | .72 | **.10** | (.50, 1.05) | .39 | 8.12 (5.21) | 7.09 (5.00) | .77 | .14 | (.55, 1.08) | .20 |
| Added Alters | 8.91 (7.07) | 6.57 (5.49) | .72 | .27 | (.41, 1.27) | .37 | 8.21 (6.51) | 6.02 (5.22) | .70 | .15 | (.43, 1.13) | .37 |
| Retained Alters | 6.21 (4.53) | 9.88 (5.25) | 1.47 | **.07** | (.99, 2.17) | .71 | 7.63 (4.82) | 10.6 (4.85) | 1.27 | .12 | (.95, 1.70) | .59 |

[1]Baseline and Follow-up means and SDs weighted from full intent-to-treat sample (N = 49) to account for non-response at follow-up.

[2]Estimates and 95% CI reported for alter count and number of components outcomes are converted to IRR to aid interpretation for non-linear models. Density estimates presented are linear model estimates.

[3]Cohen's *d* effect sizes interpreted as small (.20), medium (.50), and large (.80).

However, the average number of alters dropped from the network between baseline and follow-up was marginally lower for intervention participants compared to control participants in the SRHT-only sample, with a small effect size (p = .10, d = .39), but this association was non-significant in the full sample (p = .14). The average number of alters retained in the network between baseline and follow-up was marginally higher for intervention participants than control participants in the SRHT-only sample, with a medium to large effect size (p = .07, d = .71),

but non-significant in the full sample (p = .12). The number of new alters added to the network between baseline and follow-up did not significantly differ across treatment conditions.

## Discussion

The goal of this project was to conduct a pilot evaluation of an innovative MI-SNI using exploratory analyses to determine if the intervention was associated with changes in personal network composition and structure. Building off of previous results that demonstrated promising changes to participants' AOD use, readiness to change, and abstinence self-efficacy [34], the results presented here also demonstrate significant associations between participation in the intervention and changes in network characteristics. These findings suggest that the MI-SNI may help individuals experiencing homelessness and risky AOD use positively restructure their social networks while transitioning into supportive housing.

In terms of network composition, we found evidence from the SRHT sample that intervention participants had smaller proportions of risky network members from baseline to follow-up, namely drinking partners and network members who had any risk influence, compared to participants in the control condition. However, contrary to our expectations, we did not find any significant intervention effect on changes in the overall proportion of supportive network members. Another important finding is that intervention participants experienced more positive changes in their relationships with retained alters compared to control participants. For example, compared to control participants, those who received the intervention had a greater number of ties to alters with whom they had a drinking relationship at baseline but did not drink with in the two weeks prior to the follow-up assessment. Intervention participants also had fewer ties to alters who were rated as not being influential over their AOD use at baseline but were rated as having AOD risk characteristics at follow-up. Finally, when examining network turnover, we found that SRHT intervention participants had fewer dropped alters and more retained alters between the baseline and follow-up assessments compared to control participants, resulting in significantly denser networks with fewer components among intervention participants. The full sample analysis showed a similar result for change in components. Therefore, these results demonstrated that the MI-SNI recipients had significantly higher retention of members of their existing networks over the 3 months between assessments compared to participants in the control arm.

These findings provide preliminary evidence that intervention recipients were more likely to positively adjust their relationships with network ties they retained over the first three months after transitioning into housing compared to those who received usual case management. The findings suggest that presenting a series of network visualizations that highlighted network centrality, AOD risk, and social support may have helped MI-SNI recipients recognize both the potential for AOD risk in their networks and the network strengths that were worthy of maintaining. Although the intervention was not associated with increased social support, those who received the intervention had greater network stability and did not differ significantly in their network social support compared to those in the control condition, while reducing their overall AOD network risk overall and within retained relationships. Taken together, these findings suggest that the intervention may have triggered recipients to adjust their relationships strategically. For example, participants may have increased their awareness of risky network members, but instead of dropping them from their network, participants may have identified ways to avoid risky interactions when with these members. It is possible that combining personal network visualizations with Motivational Interviewing triggered intervention recipients to articulate active steps they could take to minimize exposure to AOD influence from network members they did not want or were not able to completely avoid. It is

possible that the MI-SNI triggered network specific "change talk" that led to behavior changes in their interactions with their network 41].

## Limitations

Although this study provides some promising results that this innovative MI-SNI design coupling Motivational Interviewing and personal network visualizations can help restructure networks in positive ways, there are several limitations worth noting. First, while our sample size is appropriate for an exploratory, small pilot study of a novel intervention approach [48], it was too small to control for factors that may have influenced the results. Also, the large number of exploratory tests run in this study is appropriate for Stage 1 behavioral therapy research development, but may have produced significant findings due to chance. Our predominantly male sample drawn from only 2 housing providers limits generalizability to other housing programs in other geographic regions with different demographic characteristics. A limitation to our tests of network change is that we were only able to collect network data immediately after the intervention period and we have no assessment of the longer-term impact of the intervention on the networks of participants. This study also relied on self-reports of network characteristics at baseline and follow-up. Due to the high respondent burden of completing personal network interviews [68,69], we had to limit our standard questions to only a few relationship characteristics. There are likely many other important relationship qualities that may be impacted by the MI-SNI intervention that we did not measure. As in other AOD use interventions, social desirability may have impacted the self-reported network AOD use outcomes, particularly for those who were invited to receive the intervention sessions and discussed their networks with MI facilitators. However, the findings showing network changes are consistent with individual-level AOD use change outcomes [34] and self-reports by egos of their alters' AOD use using a personal network approach has been found to be accurate when compared to alter self-reports [70].

Another important limitation of this study is the mixture of results that were significant for our primary sample of residents of SRHT only, the original program that contributed to the design of the intervention, and results that were significant for models based on the entire sample. These mixed findings are similar to the results of the analysis of individual level changes in AOD-related outcomes for MI-SNI recipients compared to control participants [34]. These mixed findings make it difficult to draw conclusions because there were too few SRO Housing residents (n = 13) to conduct a sub-sample analysis. Different housing programs that follow a harm reduction model operate in different ways [71] and it is possible that differences in how these two programs provide services and case management to residents impacted these mixed findings. Because of these limitations, many of the results of this exploratory analyses are preliminary and will require a larger RCT to fully test the intervention impact.

## Conclusions

Despite these limitations, these results met our initial objective to conduct a pilot test of a novel personal network-based intervention approach. The findings suggest enough promise to justify a larger RCT that enables more robust tests of hypotheses. These results provide some evidence that the intervention had an impact on intervention recipients that went beyond changes to their own personal AOD risk behavior. We believe that the findings of this pilot test suggest that coupling MI with visualizations of personal network diagrams that highlight AOD risk and support characteristics may help residents who have recently transitioned to housing to take steps to change their immediate social environment to achieve AOD use

reduction goals. These findings suggest that the intervention may have prompted actions by participants to reduce the prominence of network members who had the potential to influence their own AOD risk.

In addition to conducting a larger RCT to provide sufficient power to control for potential confounding factors, such as demographics or housing program characteristics, we recommend that future studies of this approach include a complimentary, qualitative investigation of the network change process for MI-SNI recipients compared to control participants to better understand how the intervention triggers a pattern of choices regarding which network members to retain, which to drop, and the development of relationship change strategies. This would possibly shed light on the mechanisms of network change that are triggered by coupling MI with visualizations of personal networks and key relationship characteristics related to beneficial network reconfiguration. The development of the MI-SNI and interpretation of these RCT results benefitted from qualitative data collected during beta tests of the MI-SNI interface [35] as well as other studies of formerly homeless people in substance abuse recovery [19]. Continued collection of qualitative data can provide context to better understand how people actively modify their networks to achieve behavior change outcomes.

A better understanding of the context of network change would also help assist the selection and construction of personal network measures to track changes for both control and intervention participants. We have presented one approach to measuring personal network change that met the goals of this small sample pilot test. A larger sample would enable other analytic approaches for measuring personal network change [37,39,72], including multilevel models that can test for participant—alter relationship outcomes controlling for participant, alter, and personal network characteristics while accounting for non-independence of ego-alter observations [53–55,73,74]. Although most examples of SNA informed behavior change interventions use a personal network approach, few have been rigorously tested with RCTs and longitudinal network data [30]. Therefore, this is clearly a developing field and in need of more examples to help identify best practices for measuring and testing network change. Another modification of the design used in this pilot test would be to have residents' case managers deliver the MI-SNI rather than external intervention facilitators. The visualizations resulting from the personal network interviews may help case managers understand the starting point of new residents' social environment as they transition out of homelessness and may improve their ability to understand their social challenges and recommend appropriate services.

People transitioning away from homelessness and attempting to reduce their AOD use appear to recognize the importance of the social environment in their continued AOD use. The MI-SNI may be a tool that provides them with an easy to understand personal overview of their current social environment. The four sessions that MI-SNI recipients were invited to receive may trigger them to take preliminary steps towards changing aspects of their networks while seeing tangible evidence of how these efforts impacted their networks. This progress towards social network change may encourage changes the participants' own AOD use behavior. Therefore, changing social networks may make achieving change in AOD use more attainable and may lead to greater AOD use outcomes over time. These preliminary findings suggest the need for a larger trial with a longer follow-up. Although the MI-SNI was customized for new residents of a harm reduction housing program, the results of this pilot test can also serve as promising results this intervention approach could have impact beyond the housing context. The MI-SNI intervention approach can be adapted for other populations (e.g., adolescents) and other health outcomes where social networks are influential (e.g., smoking).

## Supporting information

**S1 Appendix. CONSORT checklist.**
(PDF)

**S2 Appendix. Study protocol.** This document includes exact text describing the RCT procedures approved by the author's IRB prior to the trial beginning. The document includes both the original study plan, human subjects protection plan, and data safeguarding plan provided to the IRB in the initial ethics application as well as the final text uploaded into the human subjects review system which was discussed and approved in a full committee meeting prior to the trial starting.
(PDF)

## Acknowledgments

Thank you to the participating staff and residents at Skid Row Housing Trust and Single Room Housing Corporation without whom this research would not be possible. The authors express appreciation to David Zhang for software development, Gray Insight for assessment data collection, and Mary Lou Gilbert and Michael Bennett for intervention facilitation.

## Author Contributions

**Conceptualization:** David P. Kennedy, Karen Chan Osilla, Sarah B. Hunter, Daniela Golinelli, Joan S. Tucker.

**Data curation:** David P. Kennedy.

**Formal analysis:** David P. Kennedy, Daniela Golinelli, Ervant Maksabedian Hernandez.

**Funding acquisition:** David P. Kennedy, Karen Chan Osilla, Sarah B. Hunter, Daniela Golinelli.

**Investigation:** David P. Kennedy, Karen Chan Osilla, Sarah B. Hunter, Daniela Golinelli, Ervant Maksabedian Hernandez.

**Methodology:** David P. Kennedy, Karen Chan Osilla, Sarah B. Hunter, Daniela Golinelli, Ervant Maksabedian Hernandez, Joan S. Tucker.

**Project administration:** David P. Kennedy, Karen Chan Osilla, Sarah B. Hunter, Daniela Golinelli.

**Resources:** David P. Kennedy, Sarah B. Hunter.

**Software:** David P. Kennedy, Daniela Golinelli.

**Supervision:** David P. Kennedy, Karen Chan Osilla, Sarah B. Hunter, Daniela Golinelli.

**Validation:** David P. Kennedy.

**Visualization:** David P. Kennedy.

**Writing – original draft:** David P. Kennedy, Karen Chan Osilla, Sarah B. Hunter, Daniela Golinelli, Ervant Maksabedian Hernandez, Joan S. Tucker.

**Writing – review & editing:** David P. Kennedy, Karen Chan Osilla, Sarah B. Hunter, Daniela Golinelli, Ervant Maksabedian Hernandez, Joan S. Tucker.

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
