## [Decision Letter · Decision Letter 0]

5 Jun 2020

PONE-D-19-36073

Restructuring personal networks with a Motivational Interviewing social network intervention to assist the transition out of homelessness: a randomized control pilot study

PLOS ONE

Dear Dr. Kennedy,

Thank you for submitting your manuscript to PLOS ONE. After careful consideration, we feel that it has merit but does not fully meet PLOS ONE’s publication criteria as it currently stands. Therefore, we invite you to submit a revised version of the manuscript that addresses the points raised during the review process.

The manuscript has been evaluated by two reviewers, and their comments are available below.

The reviewers have raised a number of concerns that need attention. They request additional information on methodological aspects of the study, and revisions to the statistical analyses; moreover, they suggest edits to improve the readability of the text. 

Could you please revise the manuscript to carefully address the concerns raised?

We look forward to receiving your revised manuscript.

Kind regards,

Carmen Melatti

Associate Editor

PLOS ONE

Journal Requirements:

2. Please provide additional details regarding participant consent. In the ethics statement in the Methods and online submission information, please ensure that you have specified (1) whether consent was informed and (2) what type you obtained ( if verbal,  please specify how it was documented and witnessed).

'The authors have declared that no competing interests exist. 

We note that one or more of the authors are employed by a commercial company: RAND Corporation.

Additional Editor Comments (if provided):

Reviewers' comments:

Reviewer's Responses to Questions

**Comments to the Author**

1. Is the manuscript technically sound, and do the data support the conclusions?

Reviewer #1: Yes

Reviewer #2: Yes

2. Has the statistical analysis been performed appropriately and rigorously? 

Reviewer #1: Yes

Reviewer #2: No

3. Have the authors made all data underlying the findings in their manuscript fully available?

Reviewer #1: Yes

Reviewer #2: Yes

4. Is the manuscript presented in an intelligible fashion and written in standard English?

Reviewer #1: Yes

Reviewer #2: No

5. Review Comments to the Author

Reviewer #1: Thank you for the opportunity to review PONE-D-19-36073, entitled “Restructuring personal networks with a Motivational Interviewing social network intervention to assist the transition out of homelessness: a randomized control pilot study.” The study addresses an important problem in a novel way with an interesting intervention, and makes a worthwhile contribution. However, I have two significant concerns. First, the manuscript reads like a dissertation, with excess detail and text, which would deter readers. I recommend reducing length by 30-50%. Second, the two samples and many analyzed variables also reduces the ability to extract important results. This could be addressed by specifying primary vs. exploratory outcome variables, and by either specifying one sample as primary and the second as sensitivity analyses, or just doing an interaction by housing type. But together, these two issues make the paper overwhelming to read and digest, although it seems to be well-planned, thorough, and technically sound.

Other comments:

-Please be clearer about terms - name generator/name interpreter. At first I thought this was a coding system to list and identify contacts, but then it sounded like it meant that the participant was prompted to list names and then give details on them. This might be standard terminology for social network papers but I was not familiar with it.

-This type of network analyses simplifies friendship and human interaction by reducing it to only a few factors (users/nonusers & supportive/not) in deciding if they want to continue interactions. Much is lost in here (i.e. attraction, finding someone annoying, just “getting along” well). I acknowledge that the discussed factors are very important ones, and I see the intervention demonstrates utility. However, I think this simplification should be acknowledged in the discussion.

-Be careful in discussing decreases that are not significant as decreases.

-Be careful in discussing changes as significant in the full sample – in many cases I think you mean significant in the intervention condition using the full sample.

-Tables 4a/b (and associated text) are confusing – by the time I got here I forgot what the indicators meant and lost interest in trying to figure it out.

Reviewer #2: The authors consider a randomized control trial, specifically to test if the MI-SNI intervention is associated with longitudinal changes in personal networks, compared to subjects receiving usual care management. The topic is interesting, and the background science is compelling. However, I have some questions on the design considerations.

1. The paper is too long; for example, it took a while to read the long Introduction, and finally get to the point on what the authors are actually doing here. By making the writeup and description concise (and avoiding unnecessary details), the reading process will be more appealing. This suggestion applies to all sections.

2. Regarding sample size, the authors allude to a reference on Stage 1b pilot trial, # 55, stating considerations of 15-30 subjects in each arm. More details are necessary on what statistical tests, and effect sizes were considered to come up with that number.

3. Analyses section: From what I understand, the non-response weights were calculated using generalized boosted models, which is great, but evidence is required that this is also better than standard logistic regression for this case. It won't be hard to run and compare results from a standard logistic regression fit.

4. It was very hard, again, to find the full list of covariates (baseline outcome and demographic variables), in the middle of such long descriptions. Providing the exact list of covariables in the analyses section will allow readers to understand what actually went in the modeling.

6. PLOS authors have the option to publish the peer review history of their article (what does this mean?). If published, this will include your full peer review and any attached files.

Reviewer #1: No

Reviewer #2: No

---

## [Author Response · Author response to Decision Letter 0]

3 Aug 2020

Response to Reviewer Comments

Responses to Reviewer #1: 

1. First, the manuscript reads like a dissertation, with excess detail and text, which would deter readers. I recommend reducing length by 30-50%. 

Response: We appreciate the critique from both reviewers that the previous manuscript submission was too long. We agreed that the manuscript would benefit greatly from a significant reduction in size and scope. Therefore, we have reduced each section of the manuscript to meet the recommendations provided by the reviewers. We removed description of study details, figures, and tables that have been published elsewhere. For example, we now only provide a brief overview of the sample demographics, details about the intervention delivery, details about the programs, etc. and direct readers to publications that provide full details. This helped to reduce the length of the introduction, which was a problem noted by both reviewers. 

We also substantially reduced the number of variables described in the manuscript by removing 2 tables. The variables analyzed in those tables were the most difficult to explain and interpret. Removal of these findings did not detract from our overall discussion of results and conclusions. The manuscript length is now less than 2/3 the size of the original submission (7,078 words compared to 11,135 words and 25 pages compared to 40) with only 4 tables compared to 7 in the initial submission and only 1 figure compared to 3. 

5. Second, the two samples and many analyzed variables also reduces the ability to extract important results. This could be addressed by specifying primary vs. exploratory outcome variables, and by either specifying one sample as primary and the second as sensitivity analyses, or just doing an interaction by housing type. 

But together, these two issues make the paper overwhelming to read and digest, although it seems to be well-planned, thorough, and technically sound.

Response: We agree that the presentation of the analysis and results required editing to call greater attention to them most important findings. We addressed this according to the reviewer recommendations. We clarify that the primary sample is SRHT and the full sample is a secondary analysis. We identify the final set of analyses, those describing change in network structure and network member turn-over, as the exploratory outcome variables. We also removed the description of the variables in Tables 3a-b from the manuscript because they were the most complicated to construct and explain and the results were mostly redundant with the other results. 

We agree that using housing type as an interaction term is a logical approach. However, the sample size was not powered for an interaction between housing type and outcomes because assessing multiple housing types was not an initial research aim. 

6. Please be clearer about terms - name generator/name interpreter. At first I thought this was a coding system to list and identify contacts, but then it sounded like it meant that the participant was prompted to list names and then give details on them. This might be standard terminology for social network papers but I was not familiar with it.

Response: We appreciate the reviewer calling attention to terms that may not be clear to readers who are not familiar with social network terminology. Based on this feedback, we have re-written the section introducing this concept and clarified the terms used in a typical personal network data collection instrument. 

7. This type of network analyses simplifies friendship and human interaction by reducing it to only a few factors (users/nonusers & supportive/not) in deciding if they want to continue interactions. Much is lost in here (i.e. attraction, finding someone annoying, just “getting along” well). I acknowledge that the discussed factors are very important ones, and I see the intervention demonstrates utility. However, I think this simplification should be acknowledged in the discussion.

Response: We clarify in the limitations section that, because of the significant respondent burden in answering numerous questions in a typical personal network interview, we were unable to capture information about a wide range of important relationship characteristics. Therefore, our analysis is based on a few of the most important relationship domains that we directly targeted during our intervention development. We also add to our discussion of the benefits of qualitative analysis that an exploratory approach could help to identify key variables relevant for testing the intervention’s relationship change mechanisms in future studies. Intervention sessions are primarily directed by participants themselves who are guided by a facilitator/case manager using a Motivational Interviewing intervention approach, which stresses open ended questions and meeting participants where they want to be in their motivation to change. Providing network visualizations to participants in the intervention sessions highlights some key aspects of the network but does not restrict participants from discussing any of the other relationship characteristic that the reviewer mentions that may not be captured with our structured pre- and post-intervention instruments. 

5. Be careful in discussing decreases that are not significant as decreases.

Response: We thank the reviewer for calling attention to references to decreases in measures that were not significantly different from baseline. We have removed this language from the results description and now only describe these results as non-significant. 

6. Be careful in discussing changes as significant in the full sample – in many cases I think you mean significant in the intervention condition using the full sample.

Response: We thank the reviewer for calling attention to description of significance related to the full sample. We modified the text to avoid “significant in the full sample” and instead use the more specific language recommended by the reviewer. 

7. Tables 4a/b (and associated text) are confusing – by the time I got here I forgot what the indicators meant and lost interest in trying to figure it out.

Response: We have modified the presentation of these tables. We have re-written the associated text describing the variables in this table to simplify the language. We also re-organized the table to better match the presentation of the results. Rather than refer to these variables collectively as “network churn” variables, we now refer to them more specifically as ‘network structure” and “network member turn-over” variables. We removed the previous two tables that described complicated variables which were less important than the findings presented in these tables and more difficult to interpret. We hope that the revised presentation of these results will help readers better understand the information being conveyed in these tables. 

Responses to Reviewer #2

1. The paper is too long; for example, it took a while to read the long Introduction, and finally get to the point on what the authors are actually doing here. By making the writeup and description concise (and avoiding unnecessary details), the reading process will be more appealing. This suggestion applies to all sections.

Response: The reviewer provided the same feedback as reviewer 1 in point 1. We thoroughly revised each section according to the suggestions of both reviewers. Each section is now significantly shorter. 

 2. Regarding sample size, the authors allude to a reference on Stage 1b pilot trial, # 55, stating considerations of 15-30 subjects in each arm. More details are necessary on what statistical tests, and effect sizes were considered to come up with that number.

Response: The reference cited for this number comes from a description of the results of a series of workshops led by the National Institute of Drug Abuse. The workshops were intended to provide guidance to researchers and reviewers on the scope of different stages of intervention development research. The funding for this project comes from a grant mechanism that funds initial stages of intervention development to provide empirical pilot data to justify funding for larger clinical trials. Another objective of the funding for early stage pilot data collection and analysis is to provide the empirical data for estimating effect sizes of key variables in order to determine necessary sample size for a larger Stage 2 clinical trial. The summary of the workshop discussions to produce these recommendations does not reference any statistical tests or effect sizes used to produce this sample size range. We clarify in the text that this number range is a “rule of thumb” recommendation from NIDA rather than based on a previous statistical test. 

4. Analyses section: From what I understand, the non-response weights were calculated using generalized boosted models, which is great, but evidence is required that this is also better than standard logistic regression for this case. It won't be hard to run and compare results from a standard logistic regression fit.

Response: Our response: We selected the weighting approach based on the literature and our own prior empirical research which has shown that propensity score/non-response weights estimated with modern machine learning methods, such as boosting, are less variable and are better at reducing non-response bias than the weights obtained using the traditional logistic regression. This is, in part, due to the fact that boosting is a very flexible, non-parametric method that allows for interactions and essentially automatically selects the best model and variables that are more predictive of response. In addition to the citations we provided in the initial submission, we added the following reference to further justify our choice of the boosting approach:

Lee, B., Lessler, J., and Stuart, E.A. (2009). Improving propensity score weighting using machine learning. Statistics in Medicine 29(3): 337-346. PMCID: PMC2807890. www.ncbi.nlm.nih.gov/pubmed/19960510.

In addition, in response to the reviewer recommendation to empirically justify the choice of boosting for this case, we compared our non-response weights using boosted logistic regression to traditional logistic regression. We found that the design effect induced by boosting weights was 1.006 and the one by the logistic regression weights was 1.0182; while the difference is not large, given the modest sample size, it does impact the ability to detect significant effects. The smaller design effect for the boosting weights attests to the fact that boosting produces weights with lower variance.

We then assessed the balance between responders and non-responders, obtained using the boosting and the logistic regression weights. In this case the boosting weights yielded a better balance or, in other words, seems to be more effective in reducing non-response bias. We conducted this analysis using the R package “twang” according to the procedures in the following R documentation that we cited in our original submission:

Ridgeway G, McCaffrey D, Morral A, Griffin BA, Burgette L. (2016) Package ‘twang’. 2016. https://cran.r-project.org/web/packages/twang/twang.pdf

Below is the raw balance table output (using the “summary()” command from the R twang output) we evaluated for the boosting and logistic weighting approach comparison. 

Boosting weight output

 n.treat n.ctrl ess.treat ess.ctrl max.es mean.es max.ks max.ks.p mean.ks

unw 41 9 41.000000 9.000000 0.89225 0.266272 0.3495935 NA 0.1355014

ks.max.ATE 41 9 40.74712 8.404368 0.693059 0.257316 0.2816324 NA 0.127295

Logistic regression weight output

 n.treat n.ctrl ess.treat ess.ctrl max.es mean.es max.ks max.ks.p mean.ks

unw 41 9 41.00000 9.000000 0.89225 0.265486 0.349594 0.1355014 NA

logregw 41 9 40.28368 1.419717 0.9429 0.374439 0.624577 0.2289699 NA

As this additional statistical analyses supports the approach we took to produce weights, we have not modified the findings we present in the manuscript from our original submission, other than to reduce the number of tests described and to improve clarity. We also did not include an extensive explanation of this statistical comparison and a description of how to interpret this output in the revised manuscript. In light of the reviewers’ primary critique of the original submission, we feel it would be beyond the primary scope of the paper and would add length and complexity to the description of the analysis. If the reviewers and the editor consider this important content to include for readers, we are happy to provide this information upon request. 

5. It was very hard, again, to find the full list of covariates (baseline outcome and demographic variables), in the middle of such long descriptions. Providing the exact list of covariables in the analyses section will allow readers to understand what actually went in the modeling.

Response: We re-wrote the analysis section to better organize, simplify and present the outcome variables. We added a section listing the demographic and substance use variables included in the non-response weights. We also clarify how each model was constructed and explain that each row of the tables represents one model with one outcome variable testing for the intervention effect with only the baseline network variable as a control for the pre-post network comparison models or the network size at baseline as the controls in models of cross-wave network outcome variables.

---

## [Decision Letter · Decision Letter 1]

20 Dec 2021

Restructuring personal networks with a Motivational Interviewing social network intervention to assist the transition out of homelessness: a randomized control pilot study

PONE-D-19-36073R1

Dear Dr. Kennedy,

We’re pleased to inform you that your manuscript has been judged scientifically suitable for publication and will be formally accepted for publication once it meets all outstanding technical requirements.

Kind regards,

Jamie Males

Staff Editor

PLOS ONE

Additional Editor Comments (optional):

Reviewers' comments:

Reviewer's Responses to Questions

**Comments to the Author**

1. If the authors have adequately addressed your comments raised in a previous round of review and you feel that this manuscript is now acceptable for publication, you may indicate that here to bypass the “Comments to the Author” section, enter your conflict of interest statement in the “Confidential to Editor” section, and submit your "Accept" recommendation.

Reviewer #2: All comments have been addressed

2. Is the manuscript technically sound, and do the data support the conclusions?

Reviewer #2: (No Response)

3. Has the statistical analysis been performed appropriately and rigorously? 

Reviewer #2: (No Response)

4. Have the authors made all data underlying the findings in their manuscript fully available?

Reviewer #2: (No Response)

5. Is the manuscript presented in an intelligible fashion and written in standard English?

Reviewer #2: (No Response)

6. Review Comments to the Author

Reviewer #2: (No Response)

7. PLOS authors have the option to publish the peer review history of their article (what does this mean?). If published, this will include your full peer review and any attached files.

Reviewer #2: No

---

## [Editor Report · Acceptance letter]

13 Jan 2022

PONE-D-19-36073R1 

Restructuring personal networks with a Motivational Interviewing social network intervention to assist the transition out of homelessness: a randomized control pilot study 

Dear Dr. Kennedy:

I'm pleased to inform you that your manuscript has been deemed suitable for publication in PLOS ONE. Congratulations! Your manuscript is now with our production department. 

Kind regards, 

on behalf of

Dr Jamie Males 

Staff Editor

PLOS ONE